# Therapeutic potentials of nonpeptidic V2R agonists for partial cNDI-causing V2R mutants

**Ritsuki Kuramoto**[1◎], **Ryoji Kise**[1◎]*, **Mayu Kanno**[1◎], **Kouki Kawakami**[1¤], **Tatsuya Ikuta**[1], **Noriko Makita**[2], **Asuka Inoue**[1]*

**1** Graduate School of Pharmaceutical Sciences, Tohoku University, Sendai, Miyagi, Japan, **2** Division of Nephrology and Endocrinology, Graduate School of Medicine, The University of Tokyo, Tokyo, Japan

◎ These authors contributed equally to this work.
¤ Current address: Research Center for Advanced Science and Technology, The University of Tokyo, Tokyo, Japan
* iaska@tohoku.ac.jp (AI); ryoji.kise.d4@tohoku.ac.jp (RK)

**Data Availability Statement:** All relevant data are within the manuscript and its Supporting information files.

## Abstract

Loss-of-function mutations in the type 2 vasopressin receptor (V2R) are a major cause of congenital nephrogenic diabetes insipidus (cNDI). In the context of partial cNDI, the response to desmopressin (dDAVP) is partially, but not entirely, diminished. For those with the partial cNDI, restoration of V2R function would offer a prospective therapeutic approach. In this study, we revealed that OPC-51803 (OPC5) and its structurally related V2R agonists could functionally restore V2R mutants causing partial cNDI by inducing prolonged signal activation. The OPC5-related agonists exhibited functional selectivity by inducing signaling through the $G_s$-cAMP pathway while not recruiting β-arrestin1/2. We found that six cNDI-related V2R partial mutants (V88$^{2.53}$M, Y128$^{3.41}$S, L161$^{4.47}$P, T273$^{6.37}$M, S329$^{8.47}$R and S333$^{8.51}$del) displayed varying degrees of plasma membrane expression levels and exhibited moderately impaired signaling function. Several OPC5-related agonists induced higher cAMP responses than AVP at V2R mutants after prolonged agonist stimulation, suggesting their potential effectiveness in compensating impaired V2R-mediated function. Furthermore, docking analysis revealed that the differential interaction of agonists with L312$^{7.40}$ caused altered coordination of TM7, potentially contributing to the functional selectivity of signaling. These findings suggest that nonpeptide V2R agonists could hold promise as potential drug candidates for addressing partial cNDI.

## Introduction

Congenital nephrogenic diabetes insipidus (cNDI) is a polyuric disorder characterized by impaired water reabsorption, resulting from the diminished responsiveness of renal tubular cells to the peptidic antidiuretic hormone, arginine vasopressin (AVP). Approximately 90% of cNDI is attributed to inactivating mutations in the *AVPR2* gene, which is located on the X chromosome and encodes vasopressin V2 receptor (V2R), and the remaining 10% of cases arises from mutations in the *AQP2* (aquaporin-2) gene. To date, more than 250 putative disease-causing *AVPR2* mutations have been reported [1]. The current treatment strategy primarily focuses on symptom management through the control of urinary output and dehydration, and available options for curing the disease remain limited [2].

**Funding:** A.I. was funded by KAKENHI JP21H04791, JP21H05113 and JP21H05037 from the Japan Society for the Promotion of Science (JSPS); JPMJFR215T and JPMJMS2023 from the Japan Science and Technology Agency (JST); JP22ama121038 and P22zf0127007 from the Japan Agency for Medical Research and Development (AMED). The funders had no role in study design, data collection and analysis, decision to publish, or preparation of the manuscript.

**Competing interests:** The authors have declared that no competing interests exist.

Patients diagnosed with partial cNDI are potentially treated by the administration of V2R agonists. Diagnostic classification of cNDI is based on urine concentrating ability in response to desmopressin (dDAVP) administration and is divided into partial and complete form of cNDI [2, 3]. Whereas individuals with complete cNDI exhibit unresponsiveness to dDAVP administration, those with partial cNDI respond to dDAVP administration, leading to an increase in urine osmolality, reflecting their urinary concentrating ability [4–7]. To date, at least 13 mutations are known to cause partial cNDI. It is postulated that there would be additional V2R mutations leading to partial cNDI, owing to overlooked or uncharacterized mutations [3, 8]. Thus, since partial cNDI patients do not completely lose signaling function, drugs that can potentiate functions of V2R mutants hold promise for treatments.

Downstream signaling of V2R is mainly divided into two pathways: the $G_s$-cAMP pathway and the β-arrestin pathway. Activation of V2R leads to an increase in intracellular cAMP levels through the activation of heterotrimeric $G_s$ protein. Elevated cAMP subsequently activates protein kinase A (PKA), resulting in the phosphorylation of the water channel AQP2 [2, 9, 10]. Phosphorylated AQP2 then translocates from intracellular vesicles to the plasma membrane, thereby facilitating water reabsorption from the extracellular space. While activation of the $G_s$-cAMP signaling pathway is associated with antidiuretic effects, activation of the β-arrestin pathway can lead to receptor desensitization by internalization of the receptor [9]. Thus, desirable V2R agonists would exhibit a low β-arrestin activation capacity to segregate unwanted responses during the treatment. However, investigation of $G_s$-biased agonists for this receptor has been limited [11, 12]. Furthermore, the structural basis underlying signal bias is not well understood in V2R.

Several nonpeptidic V2R agonists have been developed as drug candidates for diseases related to abnormal urinary volume regulation. For example, OPC-51803 (hereafter designated as OPC5), WAY-151932 and fedovapagon are expected to address diseases such as central diabetes insipidus (CDI) and nocturia [13–15]. Those V2R agonists have been shown to be orally active and to exhibit antidiuretic activity in rats and dogs [13–17]. Although they are potential candidates for regulating urine volume, the pharmacological actions of these V2R agonists on individual V2R mutants, especially those associated with partial cNDI, have not been well tested.

In this study, we examined the properties of the OPC5-related V2R agonists on six partial cNDI-causing V2R mutants (V88$^{2.53}$M, Y128$^{3.41}$S, L161$^{4.47}$P, T273$^{6.37}$M, S329$^{8.47}$R and S333$^{8.51}$del (superscripts denote Ballesteros-Weinstein residue numbers [18])). The OPC5-related agonists activate cAMP signaling, but only weakly recruit β-arrestin to the receptor and show a low degree of V2R internalization. Several OPC5-related agonists were demonstrated to induce higher cAMP levels after prolonged exposure in certain V2R mutants compared to AVP. These findings suggest that nonpeptidic V2R agonists could be a potential treatment for partial cNDI.

## Materials and methods

### Reagents and plasmids

Arginine vasopressin (AVP) was purchased from Sigma-Aldrich. The OPC analogues (OPC-51803 (OPC5), OPC16b, OPC16g, OPC16j, OPC19a, OPC19b, OPC23b, OPC23d, OPC23h and OPC23i) and OPC4 (OPC41061) were kindly provided by Otsuka Pharmaceutical Company. Plasmids for the NanoBiT-β-arrestin-recruitment assay and GloSensor-based cAMP assay were previously described [19, 20]. For HiBiT-based luciferase-fragment complementation assay, human full-length V2R were N-terminally fused to a HiBiT cassette (HiBiT-V2R), which contained an interleukin 6 (IL6)-derived signal sequence followed by a HiBiT sequence

and a linker at the N terminus
(`MNSFSTSAFGPVAFSLGLLLVLPAAFPAP`<u>`VSGWRLFKKIS`</u>`GGSGGGGSG`; HiBiT tag under-lined; gene synthesized with codon optimization). Unless otherwise indicated, all the constructs were inserted into the `pCAGGS` expression plasmid vector. The V2R mutant constructs (V88[2.53]M, Y128[3.41]S, L161[4.47]P, T273[6.37]M, S329[8.47]R and S333[8.51]del) were generated by an in-house-modified QuikChange Site-Directed Mutagenesis Kit.

## Cell culture and transfection

HEK293A cells (Thermo Fisher Scientific) were maintained in Dulbecco's Modified Eagle Medium (DMEM, Nissui Pharmaceutical) supplemented with 5% fetal bovine serum (GIBCO, Thermo Fisher Scientific) and penicillin-streptomycin-glutamine (complete DMEM) at 37°C in a humidified incubator containing 5% $CO_2$. Transfection was performed by using polyethylenimine (PEI) solution (Polyethylenimine "Max", Polysciences). Typically, HEK293A cells were seeded in a 6-well culture plate at cell density of $2–3 \times 10^5$ cells/mL in 2 mL of the complete DMEM and cultured for one day A transfection solution was prepared by combining plasmid solution diluted in 100 μL of Opti-MEM (GIBCO, Thermo Fisher Scientific) and 100 μL of Opti-MEM containing 5 μL of 1 mg/mL PEI (Opti-MEM-PEI). The transfected cells were further incubated for 24 hours before being subjected to an assay.

## GloSensor-based cAMP assay

Plasmid transfection was performed in a 6-well plate with a mixture of 1 μg Glo-22F cAMP biosensor-encoding pCAGGS plasmid (gene synthesized with codon optimization by Genscript) and 200 ng of GPCR-encoding plasmid. After 24 hours incubation, the transfected cells were harvested with 0.53 mM EDTA-containing Dulbecco's-PBS (EDTA-PBS), centrifuged at 190 g for 5 min and suspended in 0.01% bovine serum albumin (BSA, fatty acid-free and protease-free grade, Serva) and 5 mM HEPES (pH 7.4)-containing Hank's Balanced Salt Solution (HBSS, assay buffer). The cells were seeded in a 96-well white culture plate (Greiner Bio-One) at a volume of 40 μL per well and added with 10 μL of 10 mM D-luciferin (FujiFilm Wako Pure Chemical) diluted in the assay buffer. After 2 hours incubation in the dark at room temperature, the plate was read for its initial luminescent count (integration time of 0.5 sec per well; Spectramax L, Molecular Devices). Then, 10 μL of 6 × test compounds serially diluted in the assay buffer were manually added. The plate was read for 30 min with an interval of 40 sec and 0.18 sec integration time at room temperature. The luminescence counts over 13–15 min after ligand addition were averaged and calculated the fold change value based on the initial count, then normalized to the response to 100 nM AVP where ligand-induced response saturated. Agonist-induced cAMP signals were fitted to a four-parameter sigmoidal concentration-response curve with the Hill slope constrained to an absolute value less than 1.5 using the following equation: $Y = Bottom + (Top − Bottom)/(1 + 10^{(LogEC50−X) \times HillSlope})$ (GraphPad Prism 10).

For the measurement of cAMP signals long after ligand stimulation, HEK293A cells were seeded in a 96-well white culture plate at cell density of $4 \times 10^5$ cells/mL in 80 μL of the Opti-MEM with penicillin-streptomycin-glutamine (PSG) and cultured for one day. Plasmid transfection was performed in a 96-well plate by adding 20 μL transfection mixture prepared by mixing 10 μL of Opti-MEM-PEI solution and 10 μL plasmid solution containing a mixture of 40 ng Glo-22F cAMP biosensor-encoding pCAGGS plasmid and 1 ng GPCR-encoding plasmid. At 4 hours after transfection, cells were treated with 5 μL of AVP or OPC compound at a final concentration of 1 μM. At 20 hours after transfection, the 96-well plate was centrifuged at 190 g for 1 min using microplate centrifuge (KUBOTA) and removed 80 μL of supernatant.

The cells were treated with 25 μL of 4 mM D-luciferin solution diluted in the assay buffer per well and centrifuged at 190 g for 1 min. After 2 hours incubation with D-luciferin in the dark at room temperature, the luminescence was measured with an interval of 1 min and 0.4 sec integration time for 5 min using Spectramax L.

## NanoBiT-β-arrestin recruitment assay

Plasmid transfection was performed in a 6-well plate with a mixture of 100 ng SmBiT-β-arrestin, 500 ng LgBiT-CAAX and 200 ng of a test V2R construct. After 24 hours incubation, the transfected cells were harvested with EDTA-PBS, centrifuged, and suspended in 2 mL of 0.01% BSA and 5 mM HEPES (pH 7.4)-containing HBSS (assay buffer). The cell suspension was dispensed in a 96-well white culture plate at a volume of 80 μL per well and added with 20 μL of 50 μM coelenterazine (Carbosynth) diluted in the assay buffer. After 2 hours incubation in the dark at room temperature, the plate was read for its baseline luminescence (SpectraMax L, Molecular Devices). Then, 20 μL of 6 × test compounds serially diluted in the assay buffer were manually added. The plate was read for 15 min with an interval of 20 sec and 0.18 sec integration time at room temperature. The luminescence counts over 13–15 min after ligand addition were averaged and calculated the fold change value based on the initial count, then normalized to the response to 100 nM AVP where ligand-induced response saturated. Agonist-induced β-arrestin signals were fitted to a four-parameter sigmoidal concentration-response curve with the Hill slope constrained to an absolute value less than 1.5 using the following equation: $Y = Bottom + (Top − Bottom)/(1 + 10^{(LogEC50−X) × HillSlope})$ (GraphPad Prism 10).

## Preparation of LgBiT

LgBiT recombinant protein was prepared as described previously [21]. Briefly, LgBiT containing the N-terminus His6 tag (GGSHHHHHHSSG), thrombin cleavage site (LVPRGS), T7 tag (HMASMTGGQQMGR), and GS linker (GGGGSGGGGS) (GenScript) was expressed in BL21 (DE3) cells and purified using Ni-NTA resin (Qiagen). The purified LgBiT protein was mixed and stored at − 80˚C until the usage.

## HiBiT-based luciferase-fragment complementation assay

HEK293A cells were seeded in a 96-well plate at cell density of $4 × 10^5$ cells/mL in 80 μL of the Opti-MEM with PSG and cultured for one day. Plasmid transfection was performed in the 96-well plate with 20 μL of transfection mixture containing PEI and 1 ng HiBiT-V2R plasmid. After 24 hours incubation, the 96-well plates were centrifuged at 190g for 1 min using microplate centrifuge and removed 75 μL of supernatant. The cells were added with 25 μL of substrate buffer consisting of 1:1000 of a recombinant LgBiT stock solution and 20 μM furimazine (Chemspace) in the assay buffer per well and centrifuged at 190 g for 1 min using microplate centrifuge. After 1 hour incubation at room temperature, the luminescent signal was measured for 5 min using SpectraMax L. For the measurement of pharmacochaperone activity, HEK293A cells were seeded in a 96-well plate at cell density of $4 × 10^5$ cells/mL in 80 μL of the Opti-MEM with PSG and cultured for one day. Plasmid transfection was performed in the 96-well plate with 20 μL of Opti-MEM-PEI solution containing a mixture of 1 ng HiBiT-V2R plasmid. After 4 hours incubation, cells were treated with 5 μL of AVP or OPC at a final concentration of 1 μM. After 20 hours, the 96-well plates were centrifuged at 190 g for 1 min using plate spin and removed 80 μL of supernatant. The cells were added with 25 μL of substrate buffer consisting of 1:1000 of a recombinant LgBiT stock solution and 20 μM furimazine in the assay buffer per well and centrifuged at 190 g for 1 min using microplate centrifuge. After

1 hour incubation in the dark at room temperature, the luminescence was measured for 5 min with an interval of 1 min and 0.4 sec integration time using SpectraMax L.

## Docking simulation

The chemical structure of the OPC16g was drawn using ChemSketch (ACD/ChemSketch, version 2018.2.1, Advanced Chemistry Development, Inc., www.acdlabs.com, 2018) and subsequently imported into ChemSketch 3D and saved as a mol2 file. The AVP-bound V2R-$G_s$ (PDB ID: 7DW9) was visualized using PyMol (Schrodinger) [22]. OPC16g was docked in the orthosteric pocket of the receptor by AutoDockFR with 50 genetic algorithm evolutions and a maximum of 20,000 evaluations [23]. The highest-scoring conformation was visualized with PyMol.

## Statistical analyses

Statistical analyses were performed using the GraphPad Prism 10 software (GraphPad). Concentration-response curves were fitted to all data by the Nonlinear Regression: Variable slope (four parameter) in the Prism 10 tool with a constraint of the Hill Slope of absolute value less than 1.5. For multiple comparison analysis, we tested statistical significance by one-way ANOVA, followed by the Dunnett's test or the Tukey's test, or the multiple t-test.

## Results

### Effects of the OPC5 analogues on functionalities of the wild-type V2R

We focused on the nonpeptidic V2R agonist OPC-51803 (OPC5) and its nine analogues (OPC16b, OPC16g, OPC16j, OPC19a, OPC19b, OPC23b, OPC23d, OPC23h and OPC23i) (Fig 1A and S1 Table). OPC5 and its analogues used in this study were described previously [24, 25] and, in this study, they are collectively referred to as OPC5 analogues.

To elucidate signaling properties induced by the OPC5 analogues, we evaluated activation of the $G_s$ and β-arrestin pathways upon agonist stimulation. Previously, activation of the β-arrestin pathway was not tested for OPC5 analogues as compared to the $G_s$-cAMP pathway [24]. First, we measured intracellular cAMP levels following stimulation with the OPC5 analogues to evaluate the activation of the $G_s$ pathway. AVP, an endogenous agonist for V2R, was used as the reference ligand, inducing robust activation of both G protein and β-arrestin pathways (S1 Fig). All ten OPC5 analogues elevated intracellular cAMP levels with efficacy comparable to AVP, but with significantly lower potency (Fig 1B, S1A Fig and S2 Table). The degree of reduced affinity of the OPC5 analogues for V2R compared to AVP is in line with previous studies [11, 24, 25]. Secondly, we examined the recruitment of each of the β-arrestin subtypes, β-arrestin1 and β-arrestin2, in response to agonist stimulation. While AVP stimulation induced a substantial recruitment of β-arrestin to the plasma membrane, the OPC5 analogues induced only minimal β-arrestin recruitment (Fig 1C–1E, S1 Fig and S2 Table). In addition to OPC5 and OPC23h, for which weakened β-arrestin signaling was reported [7, 11], we found that the other OPC5 analogues have a limited β-arrestin activation capacity. Although the Glo-Sensor-cAMP assay may not adequately assess efficacy of agonists due to its high sensitivity, these results suggest that the OPC5 analogues have a functional selectivity for the $G_s$-cAMP pathway over the β-arrestin pathway.

Next, we examined receptor internalization activity of the OPC5 analogues. To measure cell-surface expression of V2R, we used HiBiT-based luciferase-fragment complementation assay (Fig 2A). This method measures the luminescence induced by the reconstitution of two luciferase fragments, LgBiT and HiBiT. By adding membrane-impermeable LgBiT protein to

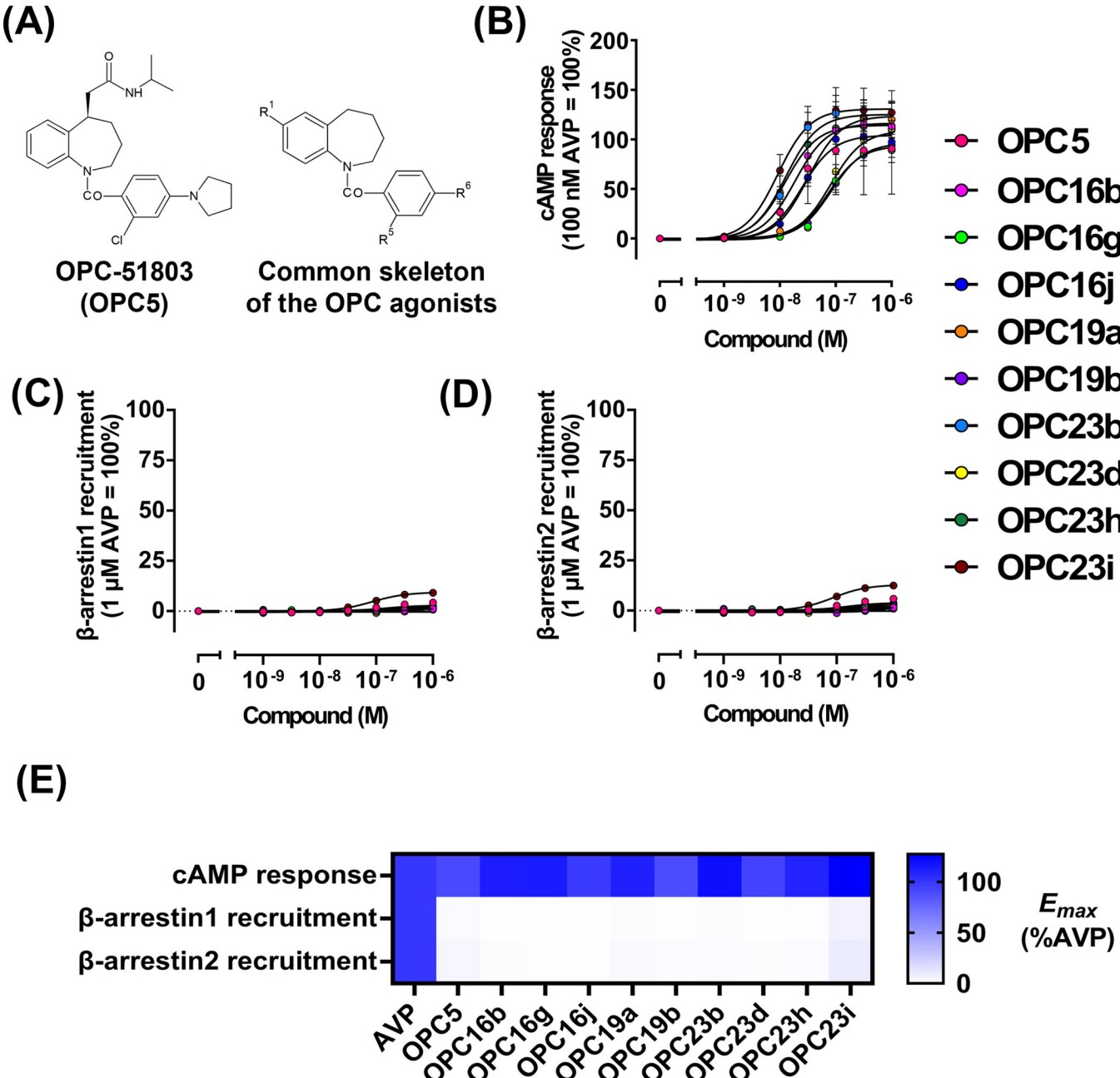

**Fig 1. The OPC5 analogues activate G$_s$-cAMP pathway but not β-arrestin pathway.** (A) Structure of OPC5 and common skeleton of OPC5 analogues. (B) Concentration-response curve for cAMP accumulation upon the OPC analogues stimulation. For each experiment, ligand-induced cAMP responses were normalized to the response induced by 100 nM AVP. Symbols and error bars are mean and SEM, respectively, of three independent experiments with each performed in duplicate. AVP, OPC5, OPC16b, OPC16g, OPC16j, OPC19a, OPC19b, OPC23b, OPC23d, OPC23h and OPC23i are colored in black, red, magenta, light green, blue, orange, purple, cyan, yellow, green and brown, respectively. (C and D) Concentration-response curve for β-arrestin1 (C) and β-arrestin2 (D) recruitment upon the OPC5 analogues stimulation. For each experiment, ligand-induced β-arrestin recruitment was normalized to the response induced by 1 μM AVP. Symbols and error bars are mean and SEM, respectively, of three independent experiments with each performed in duplicate. (E) Heatmap representation of cAMP response and β-arrestin1/2 recruitment induced by AVP and the OPC analogues. Data are presented as $E_{max}$ of each experiment and ligand-induced cAMP responses and β-arrestin1/2 recruitment were normalized to the response induced by 100 nM or 1 μM AVP, respectively.

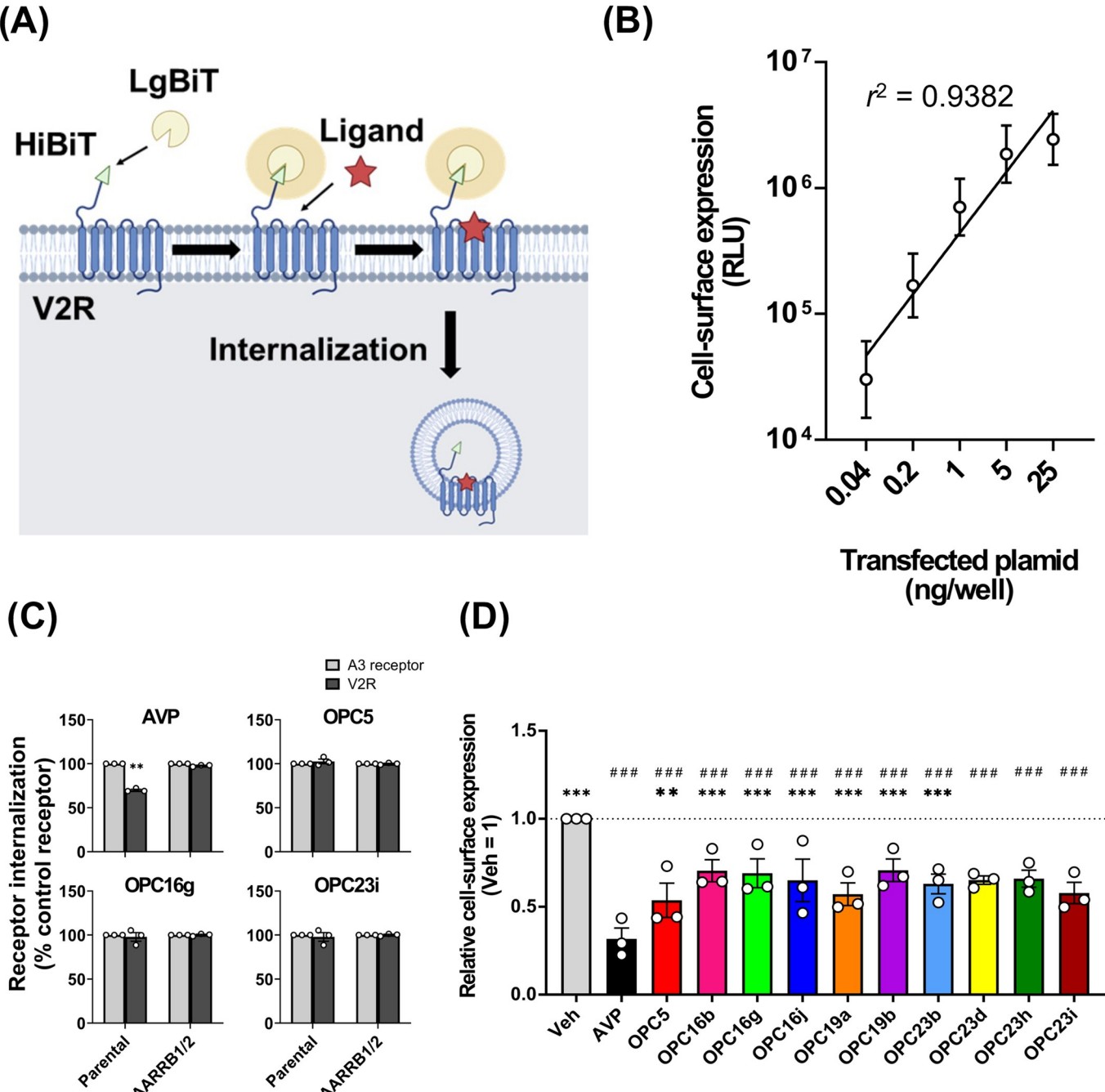

**Fig 2. OPC5 analogues have less V2R internalizing activity than AVP.** (A) Schematic representation of the HiBiT-based luciferase-fragment complementation assay. The amount of plasma membrane receptor was assessed by reconstituting the luciferase with membrane-impermeable LgBiT. Agonist-induced internalization of the receptor results in a decline of luminescence signals. (B) Measurement of cell-surface expression levels with a titrated amount of wild-type V2R (0.04–25 ng). (C) Measurement of cell-surface expression levels of wild-type V2R. For each experiment, cell-surface expression levels were normalized to the control receptor. Bars and error bars are mean and SEM, respectively, of three independent experiments with each performed in duplicate. For the statistical analyses, data were analyzed by paired t-test. *$P < 0.05$, **$P < 0.01$, ***$P < 0.001$. (D) Measurement of cell-surface expression levels of wild-type V2R upon AVP or the OPC5 analogues stimulation for 20 hours. For each experiment, cell-surface expression levels were normalized to the expression levels upon vehicle stimulation. Bars and error bars are mean and SEM, respectively, of three independent experiments with each performed in duplicate. For the statistical analyses, data were analyzed by one-way ANOVA followed by the Dunnett's test for multiple comparison analysis. *$P < 0.05$, **$P < 0.01$, ***$P < 0.001$ vs. AVP; # $P < 0.05$, ##$P < 0.01$, ###$P < 0.001$ vs. vehicle.

the cell media, only the N-terminally HiBiT-labeled V2Rs located on the plasma membrane are detected. The luminescence signals correlated with the amount of transfected plasmid, indicating the feasibility of the assay to evaluate plasma membrane expression levels (Fig 2B). We found that AVP induced significant receptor internalization, but the OPC5 analogues induced negligible internalization (Fig 2C). AVP-induced receptor internalization was abolished in β-arrestin1 and β-arrestin2 double-deficient (ΔARRB1/2) HEK293A cells, indicating that V2R internalization is β-arrestin dependent. Next, we observed a decrease in plasma membrane expression levels of V2R at 20 hours after agonist stimulation (Fig 2D). Notably, OPC5 analogues induced V2R internalization at this later time point, although to a lesser extent compared to AVP stimulation. Taken together, the OPC5 analogues exhibit weaker receptor internalizing activity compared to AVP.

## Effects of the OPC5 analogues on functionalities of the cNDI-causing V2R mutants

To assess the potential for restoring the function of V2R mutants, we attempted to characterize the functional defects in individual V2R mutants. The molecular mechanisms by which disease-causing mutations cause loss of function of the V2R is divided into three types: impairment in biosynthesis process of V2R, mislocalization of V2R caused by the retention of misfolded V2R in the endoplasmic reticulum (ER) and the defects in signaling function [26]. Moreover, it is believed that these abnormalities contribute to the development of cNDI in a combined manner [3]. As causative mutations of partial cNDI, we selected six V2R mutations (V88$^{2.53}$M, Y128$^{3.41}$S, L161$^{4.47}$P, T273$^{6.37}$M, S329$^{8.47}$R and S333$^{8.51}$del) (S2A Fig). We first examined the plasma membrane expression levels for these V2R mutants. As a result, V2R mutants other than V88$^{2.53}$M had reduced plasma membrane expression compared to wild-type V2R (S2B Fig). Although V88$^{2.53}$M failed to show significant differences ($P = 0.053$), a trend toward decreased membrane expression was observed. Notably, the Y128$^{3.41}$S, L161$^{4.47}$P, and T273$^{6.37}$M mutants displayed only a minimal amount of the receptor expressed at the plasma membrane (S2B Fig). To our knowledge, our result is the first report of the reduced expression levels of the L161$^{4.47}$P mutant.

To determine the impact of each mutation on its signaling function, we measured cAMP production in response to AVP. All six V2R mutants tested required higher concentrations of AVP than the wild-type V2R for an increase in intracellular cAMP levels (Fig 3A and 3B). The observation that the response to AVP is not completely abolished in these mutants is consistent with the classification of these mutants as the cause of partial cNDI. In contrast to the higher EC$_{50}$ values for AVP in these mutants, cAMP response levels immediately after AVP stimulation were not severely impaired (Fig 3C). However, at a later time point following AVP stimulation, intracellular cAMP levels were markedly lower compared to the wild-type V2R, suggesting a weakened capacity for signal activation to sustain elevated cAMP levels in these mutants (Fig 3D). The reduced affinities and the decline in cAMP levels at a later time point in the mutants are likely attributable, at least in part, to the reduced quantity of receptors on the plasma membrane and a faster depletion of the receptor pool on the plasma membrane.

Because restoration of plasma membrane expression of V2R mutants is one promising approach to improve the receptor function, we investigated the pharmacological chaperone activity of the OPC5 analogues. In general, antagonists, but not agonists, exhibit pharmacochaperone activity to the receptors [27]. To date, several V2R antagonists including OPC3 (OPC31260, mozavaptan) and OPC4 (OPC41061, tolvaptan) are reported to restore the plasma membrane expression of V2R mutants [6]. Moreover, V2R agonists have also been reported to have pharmacological chaperone activity, albeit with lower efficiency compared to

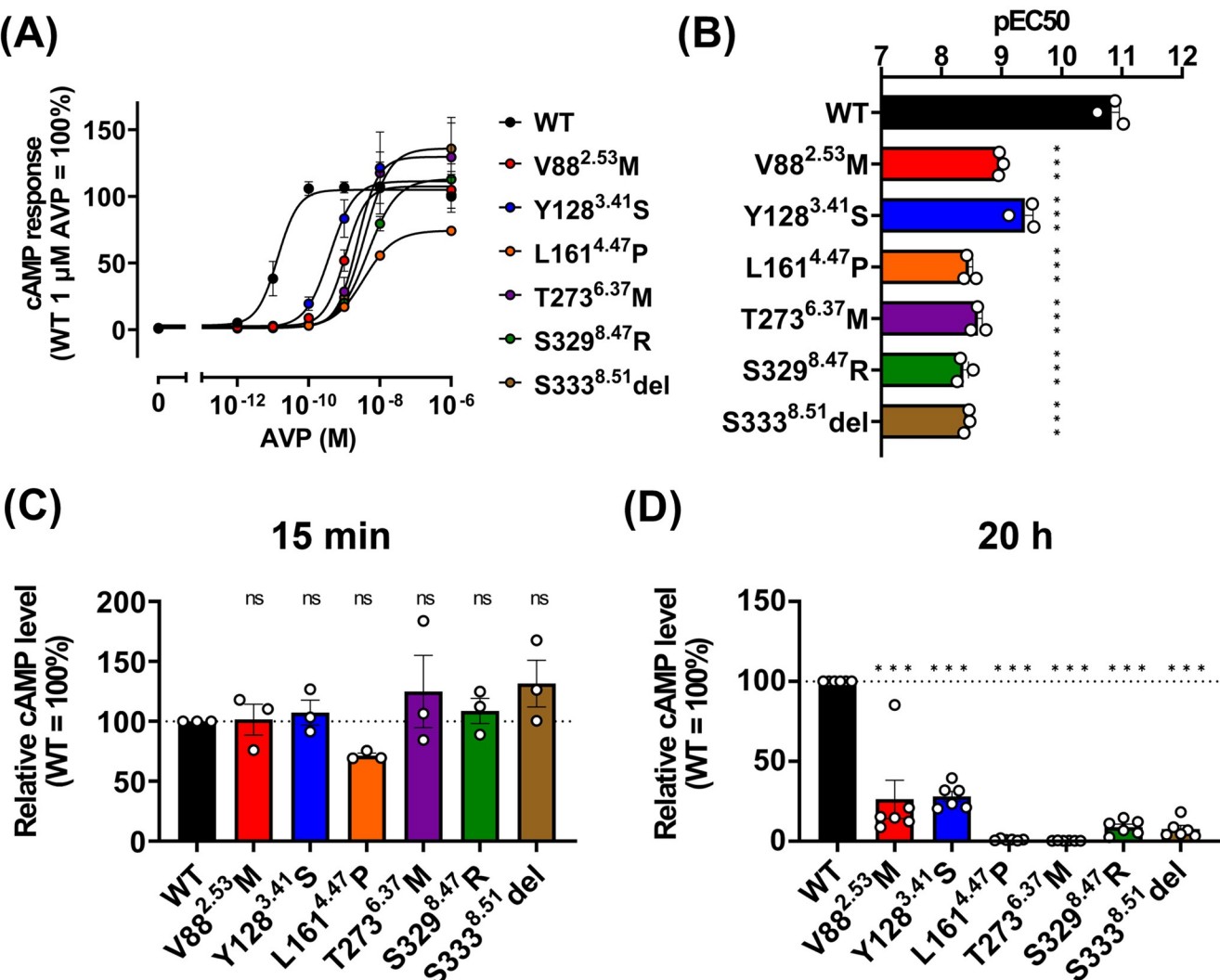

**Fig 3. V2R mutants impair prolonged increase of cAMP levels.** (A) Concentration-response curve for cAMP accumulation induced by wild-type and the six mutant V2Rs activation upon AVP stimulation. For each experiment, ligand-induced cAMP responses were normalized to the response induced by 1 μM AVP at wild-type V2R. WT, V88[2.53]M, Y128[3.41]S, L161[4.47]P, T273[6.37]M, S329[8.47]R and S333[8.51]del are colored in red, blue, orange, purple, green and brown, respectively. Symbols and error bars are mean and SEM, respectively, of three independent experiments with each performed in duplicate. (B) pEC$_{50}$ values for wild-type and six mutant V2Rs for AVP. Symbols and error bars are mean and SEM, respectively, of three independent experiments with each performed in duplicate. ***$P < 0.001$ vs. WT. (C-D) Relative cAMP levels for wild-type and the six mutant V2Rs after 15 min (C) and 20 hours (D) stimulation of AVP. Bars and error bars are mean and SEM, respectively, of three independent experiments with each performed in duplicate. ***$P < 0.001$ vs. WT.

antagonists [7, 11]. Thus, we examined the impact of OPC5 agonists on the plasma membrane expression of cNDI-causing V2R mutants. As previously demonstrated, treatment with the antagonists OPC4 restored plasma membrane expression of the Y128[3.41]S, S329[8.47]R, and S333[8.51]del mutants (S2C Fig). Among OPC5 analogues, OPC16g and OPC23h elevated plasma membrane expression of the V88[2.53]M mutant, while OPC16j elevated plasma membrane expression of the Y128[3.41]S and S329[8.47]R mutants. Taken together, we observed that some OPC5 analogues exhibit pharmacological chaperone activity, but their effects were much smaller than those of OPC4.

We next evaluated the therapeutic potential of OPC5 analogues for cNDI-causing V2R mutants. The impairment of sustained activation of cAMP pathway as well as reduced cell surface expression is a shared characteristic among the six tested V2R mutants upon AVP stimulation (Fig 3D and S2B Fig). Thus, we opted to examine whether the OPC5 analogues could restore the attenuation of the sustained cAMP signaling. After a 20-hour incubation, intracellular cAMP levels were found to be higher in several, but not all, conditions of the OPC5 analogues treatment as compared to AVP (Fig 4). The elevation in cAMP levels largely correlated with the potency of the OPC5 analogues on V2R, with the highest effects seen with the high affinity agonists OPC16g and OPC23i (S2 Table). As expected, although it increased the cell-surface expression level of V2R mutants (S2C Fig), treatment of OPC4 failed to induce cAMP accumulation (S2D Fig). Interestingly, the pattern of changes in cAMP levels correlates with the plasma membrane expression levels of each mutant. While the Y128$^{3.41}$S, L161$^{4.47}$P, and T273$^{6.37}$M mutants with low membrane expression levels exhibited increased cAMP levels in response to OPC5 analogs compared to AVP, the V88$^{2.53}$M, S329$^{8.47}$R, and S333$^{8.51}$del mutants with modest expression levels demonstrated lower cAMP levels upon stimulation of some OPC5 analogues relative to that of AVP. This indicates the OPC5 analogues are more effective against mutants with low expression levels. In fact, several OPC5 analogues induced cAMP elevation in the Y128$^{3.41}$S mutant to levels comparable to those observed in the wild-type V2R upon AVP stimulation (S3 Fig). Considering that high affinity correlates with degree of recovery, further development of higher affinity V2R agonists may lead to recovery of mutant function.

## Docking simulation of the OPC5 analogue

To gain a deeper understanding of the molecular underpinnings behind the differences in pharmacological effects between OPC5 analogues and AVP, we conducted docking simulations to investigate the mechanisms of ligand recognition by V2R. OPC16g was selected as a representative OPC5 analogue due to its high cAMP signal activity (Fig 1B and S2 Table). Docking simulations revealed that, as in the case of previously reported AVP-bound structure, residues corresponding to the six partial cNDI-causing V2R mutations were not directly involved in ligand recognition, as demonstrated by the OPC16g-docked V2R structure (Fig 5A) [22]. In addition, the positions of these six mutations do not engage in the interaction with either Gα or β-arrestin, except for the S329$^{8.47}$, which contacts the C-terminus of Gα [22, 28, 29]. Thus, these data support that the attenuated cAMP responses in the six partial cNDI-causing V2R mutants are attributable to impaired plasma membrane expression or signaling function rather than the impairment of ligand recognition.

A close inspection of the AVP-bound and OPC16g-docked structure suggested that the hydrophobic cleft located at the deep ligand binding pocket serves as a common mechanism to recognize the agonists. In the AVP-bound V2R structure, the side chains of Tyr2$^{AVP}$ and Phe3$^{AVP}$ extend toward the hydrophobic cavity at the bottom of the ligand-binding pocket, which is comprised of the amino acids including M120$^{3.33}$, M123$^{3.36}$, Y205$^{5.38}$, V206$^{5.39}$, I209$^{5.42}$, F287$^{6.51}$, F288$^{6.52}$ and Q291$^{6.55}$ [22, 28]. The docked structure of OPC16g-V2R revealed that the hydrophobic benzazepine scaffold contacted with M120$^{3.33}$, M123$^{3.36}$, I209$^{5.42}$, F287$^{6.51}$ and F288$^{6.52}$ (Fig 5B). F287$^{6.51}$ and F288$^{6.52}$ are residues located just above W284$^{6.48}$, which acts as a toggle switch for conformational changes during GPCR activation. From the studies of the active structure of V2R and its closely related oxytocin receptor (OTR), it is proposed that the downward movement of the side chain of F$^{6.51}$ contributes to the propagation of conformational changes through W$^{6.48}$ [28, 30]. Consistently, the importance of hydrophobic interactions is underscored by the fact that the alanine substitution of F287$^{6.51}$

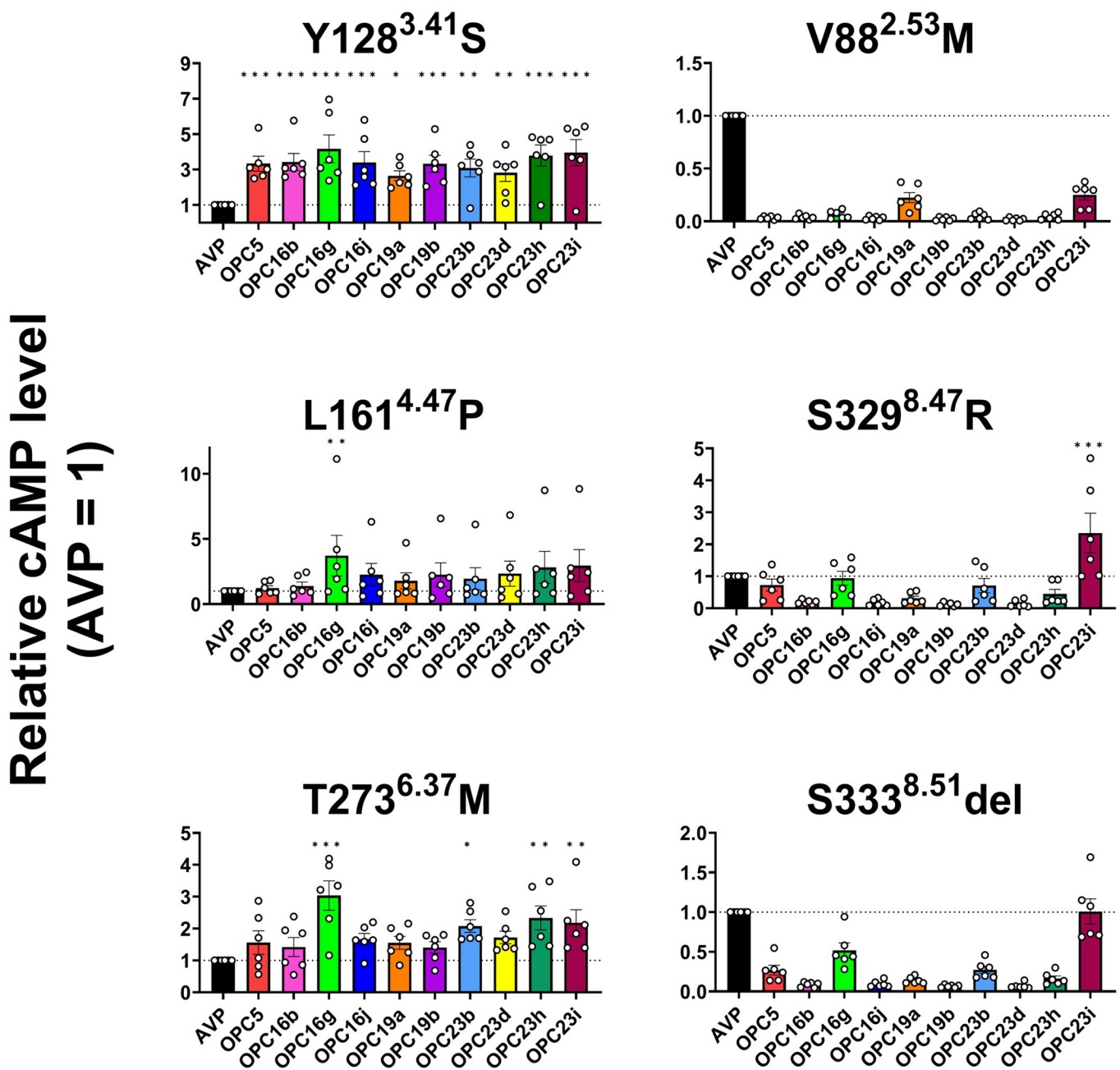

**Fig 4. OPC5 analogues cause sustained cAMP signaling in V2R mutants with low expression levels.** Measurement of cAMP levels of wild-type and the six mutant V2Rs upon AVP and OPC5 analogues stimulation for 20 hours. For each experiment, ligand-induced cAMP responses were normalized to the response induced by AVP. Bars and error bars are mean and SEM, respectively, of six independent experiments with each performed duplicate. Data were analyzed by one-way ANOVA followed by the Dunnett's test for multiple comparison analysis. *$P < 0.05$, **$P < 0.01$, ***$P < 0.001$ vs. AVP.

and F288[6.52] impaired both the cell surface expression and the affinity of AVP for V2R [22]. Taken together, OPC5 analogues induce receptor activation likely through hydrophobic interactions similar to those observed with AVP.

In contrast, unlike AVP, OPC16g does not form a hydrogen bond with the main chain of L312[7.40], which is a unique feature observed in the active conformation in V2R as well as

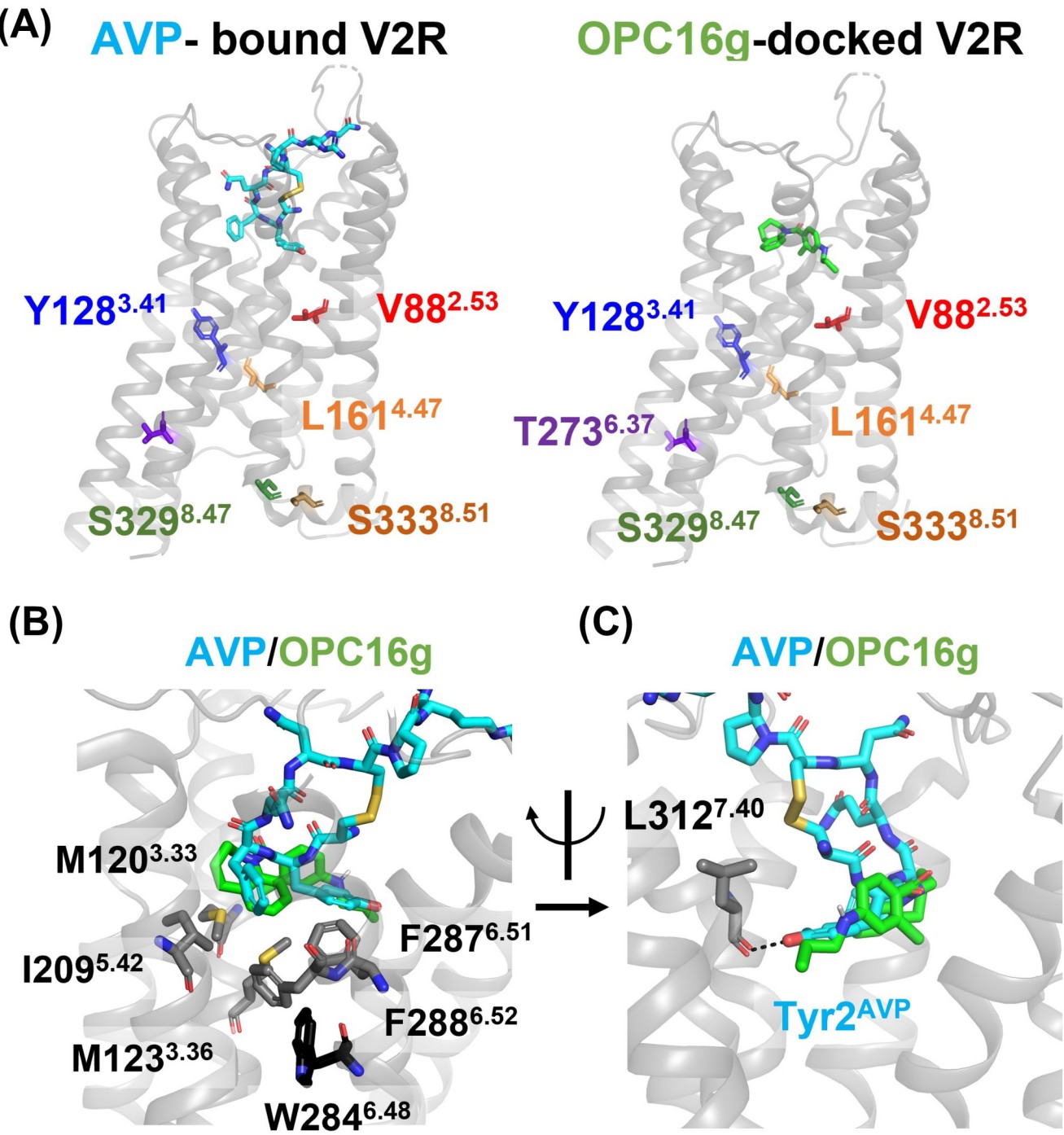

**Fig 5. Structural comparison of the AVP-bound V2R and OPC16g-docked V2R structure.** (A) Overall structure of the AVP-bound V2R-G$_s$ structure (PDB ID: 7DW9) (left panel) and OPC16g-docked V2R structure (right panel). V2R, AVP and OPC16g are colored in gray, cyan and green, respectively. V88$^{2.53}$, Y128$^{3.41}$, L161$^{4.47}$, T273$^{6.37}$, S329$^{8.47}$ and S333$^{8.51}$ are colored in red, blue, orange, purple, green and brown, respectively. (B and C) Conformational comparison of ligand binding pocket of AVP-bound V2R and OPC16g-docked V2R. Residues consisting of a hydrophobic cleft (B) and the hydrogen bond with L312$^{7.40}$ of V2R (C) were shown as sticks.

OTR (Fig 5C) [22, 30, 31]. Considering that an AVP analogue, which lacks the ability to form hydrogen bonds due to the substitution of Tyr2 with Phe2, shows a reduced binding affinity for V2R [31, 32], the absence of the interaction could partially account for why OPC16g has a lower affinity than AVP. Furthermore, since the conformational changes leading to GPCR activation are mainly transmitted by the interplay between TM3, TM6, and TM7 [33, 34], the ligand-induced rearrangement of TM7 mediated by the interaction of L312$^{7.40}$ can alter the signaling function. These structural differences in ligand recognition could explain the differences in functional selectivity of signals between OPC5 analogues and AVP.

## Discussion

This study investigated the pharmacological activity of non-peptidic V2R agonists against partial cNDI-associated V2R mutants. The results suggest that the OPC5 analogues have an advantage over AVP in terms of activating the mutant V2Rs in a prolonged time period, albeit the OPC5 analogues exhibit a lower affinity than AVP. One of the possible reasons for the prolonged activation by the OPC5 analogues is their lower internalization activity. Because activation of β-arrestin signaling leads to receptor desensitization, G-protein-biased agonists that selectively activate the G$_s$-cAMP pathway rather than the β-arrestin pathway would be particularly beneficial in V2R mutants that have low plasma membrane expression. Although low β-arrestin activity can be advantageous of prolonged cAMP signaling for OPC5 analogues, low β-arrestin activity also leads to reduced cAMP signaling originating from the endosome. Given that β-arrestin-mediated endosomal signaling is reported to contribute to the phosphorylation of AQP2 [35], further studies are needed to determine whether treatment with OPC5 analogues causes sufficient plasma membrane translocation of AQP2. Taken together, our results indicate that non-peptidic agonists could induce a higher cAMP response than peptidic agonists for partial cNDI-related V2R mutants with low plasma membrane expression levels, like the Y128$^{3.41}$S, L161$^{4.47}$P and T273$^{6.37}$M mutants.

Precision medicine has been an important strategy in the treatment of diseases whose pathophysiology is altered depending on mutations [36–38]. As mentioned earlier, the mechanisms that lead to functional loss of V2R can be divided into three categories [3]. Consistent with previous reports [6, 7, 11], our study suggests that functional loss of the Y128$^{3.41}$S mutant is mainly due to defects in translocation from ER to the plasma membrane. Considering the efficacy of the OPC5 analogues for the Y128$^{3.41}$S mutant (Fig 4 and S3 Fig), it is reasonable to assume that the OPC5 analogues may also effectively enhance cAMP signaling in other localization-defective mutants such as G201$^{5.34}$D [26]. In contrast, a loss of function of the S329$^{8.47}$R mutant is likely caused by defects in signaling function and may not be a promising target for our OPC5 analogue-based approach because the OPC5 analogues do not alleviate the signaling deficiency. In mutants with defective signaling activity, activation of G$_s$ signaling pathway via a route other than V2R may serve as an effective strategy. Together, the OPC5 analogue-based approach may become beneficial to patients with partial cNDI caused by a certain type of V2R mutations, although further in-depth studies are required in future.

Our result of docking simulation provides clues for the rational design of the signal properties of the compound. In the OPC16g-docked structure, a hydrogen bond between the ligand and the main chain of L312$^{7.40}$ was not observed. Based on the activation of V2R upon OPC16g and [Phe$^2$]AVP stimulation, a hydrogen bond to the main chain of L312$^{7.40}$ is dispensable for agonist activity, but the lack of a hydrogen bond significantly reduces the affinity for V2R [31, 32]. Thus, introducing a substituent containing a hydroxyl group at the R6

position of the OPC5 analogue may increase in affinity for V2R. In addition, a hydrogen bond with L312[7.40] induces a unique rearrangement of TM7 by pulling TM7 toward the ligand-binding pocket in V2R and OTR [31, 34]. In the course of GPCR activation, sequential conformational changes through microswitches such as the CWxP, the PIF, and the NPxxY motifs lead to the substantial outward movement of TM6, which is important for the binding of transducer protein. Although L312[7.40] is not likely to be directly involved in the propagation of these conformational changes, it may influence the signal property by affecting the relay of interactions between TMs. In fact, at least for the closely related receptor, OTR, the hydrogen bond at L[7.40] is important in determining whether the receptor undergoes full or partial activation [30]. With respect to signaling bias, the interaction between Y[7.53] and R[3.50] determines the balance between G protein signaling and β-arrestin signaling in many GPCRs [34]. Although the structures of V2R-transducer complex have been solved, the mechanism of signaling bias is not clear because the overall structure of V2R is not significantly different between G-protein-bound and β-arrestin-bound structures [22, 28, 29]. Since the OPC16g without β-arrestin activity does not interact with L312[7.40], L312[7.40] may serve as a determinant of biased signaling.

Aside from the potential influence of the hydrogen bond-mediated interaction with L312[7.40] on potency and signal property, the differential coordination of TM7 by the ligand's substructure facing L312[7.40] is likely to determine whether the compound functions as an agonist or antagonist. In fact, OPC5 differs from its closely related antagonists OPC3 and OPC4 in that it lacks the bulky methylbenzamide at the R6 position facing L312[7.40], suggesting that the arrangement around L312[7.40] is important for the antagonist-to-agonist conversion [24, 39]. In addition, similar conversion from antagonist to agonist has been reported for the compounds of a benzodiazepine scaffold [14]. In those compounds, the smaller substituents on the opposite side of the hydrophobic ring structure contribute to the conversion of antagonist activity to agonist activity. In the course of the development of the OPC5 analogues, it was demonstrated that the agonist efficacy, but not the affinity, was altered by the substituents facing L312[7.40], further supporting the importance of the coordination of TM7 through the interaction with L312[7.40] in determining the signal property [24]. These results provide a valuable starting point for the development of rational compounds. Future investigations into the mechanisms determining biased signaling and pharmacological chaperone activity will lead to the development of even more effective V2R agonists.

## Supporting information

**S1 Fig. AVP activates both $G_s$-cAMP and β-arrestin pathways.** (A) Concentration-response curve for cAMP accumulation upon AVP stimulation. For each experiment, ligand-induced cAMP responses were normalized to the response induced by 100 nM AVP. Symbols and error bars are mean and SEM, respectively, of three independent experiments with each performed in duplicate. (B and C) Concentration-response curve for β-arrestin1 (B) and β-arrestin2 (C) recruitment upon AVP stimulation. For each experiment, ligand-induced cAMP responses were normalized to the response induced by 1 μM AVP. Symbols and error bars are mean and SEM, respectively, of three independent experiments with each performed in duplicate.
(PDF)

**S2 Fig. Some OPC5 analogues possess marginal pharmacochaperone activity.** (A) Snake plot of human V2R. The residues altered in the mutants used in this study are colored in yellow. V88[2.53], Y128[3.41], L161[4.47], T273[6.37], S329[8.47] and S333[8.51] are colored in red, blue, orange, purple, green and brown, respectively. (B) Measurement of cell-surface expression of wild-

type and the six mutant V2Rs. For each experiment, cell-surface expression levels were normalized to the expression levels of wild-type V2R. Bars and error bars are mean and SEM, respectively, of three independent experiments with each performed in duplicate. $^*P < 0.05$, $^{**}P < 0.01$, $^{***}P < 0.001$ vs. WT. (C) Measurement of cell-surface expression of the six mutant V2Rs upon OPC analogues stimulation for 20 hours. For each experiment, cell-surface expression levels were normalized to the expression levels upon vehicle stimulation. Bars and error bars are mean and SEM, respectively, of four independent experiments with each performed in duplicate. For the statistical analyses, Data were analyzed by one-way ANOVA followed by the Dunnett's test for multiple comparison analysis. $^*P < 0.05$, $^{**}P < 0.01$ vs. vehicle. (D) Measurement of cAMP levels of the six mutant V2Rs upon OPC4 stimulation for 20 hours. For each experiment, ligand-induced cAMP responses were normalized to the response induced by vehicle stimulation. Bars and error bars are mean and SEM, respectively, of six independent experiments with each performed in duplicate. Data were analyzed by unpaired t-test $^*P < 0.05$, $^{**}P < 0.01$, $^{***}P < 0.001$ vs. vehicle.
(PDF)

**S3 Fig. OPC5 analogues induce cAMP elevation in the six mutant V2Rs comparable to wild-type V2R upon AVP stimulation.** Measurement of cAMP levels of wild-type and the six mutant V2Rs upon AVP and OPC analogues stimulation for 20 hours. For each experiment, ligand-induced cAMP responses were normalized to the response induced by AVP at wild-type V2R. Bars and error bars are mean and SEM, respectively, of six independent experiments with each performed in duplicate. Data were analyzed by one-way ANOVA followed by the Dunnett's test for multiple comparison analysis. $^*P < 0.05$, $^{**}P < 0.01$, $^{***}P < 0.001$ vs. AVP.
(PDF)

**S1 Table. Structure of OPC agonists.**
(PDF)

**S2 Table. pEC$_{50}$ values of cAMP response and β-arrestin1/2 recruitment induced by V2R activation upon AVP and OPC agonists.**
(PDF)

**S1 Data.**
(XLSX)

## Acknowledgments

We thank Ayaki Saito and other members of the Inoue laboratory for critical reading and editing of the manuscript. We thank Dr. Hiroyuki Fujiki at Otsuka Pharmaceutical for the OPC5 agonists.

## Author Contributions

**Conceptualization:** Ritsuki Kuramoto, Ryoji Kise, Asuka Inoue.

**Funding acquisition:** Asuka Inoue.

**Investigation:** Ritsuki Kuramoto, Mayu Kanno, Kouki Kawakami, Tatsuya Ikuta, Asuka Inoue.

**Methodology:** Ryoji Kise.

**Resources:** Noriko Makita.

**Supervision:** Asuka Inoue.

**Writing – original draft:** Ritsuki Kuramoto, Ryoji Kise, Asuka Inoue.

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
