## [Decision Letter · Decision Letter 0]

10 Jan 2024

PONE-D-23-41336Therapeutic potentials of nonpeptidic V2R agonists for partial cNDI-causing V2R mutantsPLOS ONE

Dear Dr. Inoue,

Thank you for submitting your manuscript to PLOS ONE. After careful consideration, we feel that it has merit but does not fully meet PLOS ONE’s publication criteria as it currently stands. Therefore, we invite you to submit a revised version of the manuscript that addresses the points raised during the review process. As you will notice, the comments from the reviewers are straightforward to address, and hopefully, you can address them through textual revision without additional experiments.

We look forward to receiving your revised manuscript.

Kind regards,

Arun Shukla

Academic Editor

PLOS ONE

Journal Requirements:

3. Thank you for stating the following financial disclosure: "A.I. was funded by KAKENHI JP21H04791, JP21H05113 and JP21H05037 from the Japan Society for the Promotion of Science (JSPS); JPMJFR215T and JPMJMS2023 from the Japan Science and Technology Agency (JST)."

4. Thank you for stating the following in the Acknowledgments Section of your manuscript: "We thank Ayaki Saito and other members of the Inoue laboratory for critical reading and editing of the manuscript. We thank Dr. Hiroyuki Fujiki at Otsuka Pharmaceutical for the OPC5 agonists. A.I. was funded by KAKENHI JP21H04791, JP21H05113 and JP21H05037 from the Japan Society for the Promotion of Science (JSPS); JPMJFR215T and JPMJMS2023 from the Japan Science and Technology Agency (JST)."

Please remove any funding-related text from the manuscript and let us know how you would like to update your Funding Statement. Currently, your Funding Statement reads as follows: "A.I. was funded by KAKENHI JP21H04791, JP21H05113 and JP21H05037 from the Japan Society for the Promotion of Science (JSPS); JPMJFR215T and JPMJMS2023 from the Japan Science and Technology Agency (JST)."

5. We note that your Data Availability Statement is currently as follows: "All relevant data are within the manuscript and its Supporting Information files."

Reviewers' comments:

Reviewer's Responses to Questions

**Comments to the Author**

1. Is the manuscript technically sound, and do the data support the conclusions?

Reviewer #1: Partly

Reviewer #2: Yes

Reviewer #3: Yes

2. Has the statistical analysis been performed appropriately and rigorously? 

Reviewer #1: Yes

Reviewer #2: Yes

Reviewer #3: Yes

3. Have the authors made all data underlying the findings in their manuscript fully available?

Reviewer #1: Yes

Reviewer #2: Yes

Reviewer #3: Yes

4. Is the manuscript presented in an intelligible fashion and written in standard English?

Reviewer #1: Yes

Reviewer #2: Yes

Reviewer #3: Yes

5. Review Comments to the Author

Reviewer #1: Kuramoto et al investigate the effect of nonpeptidic V2R agonists on V2R mutants that cause congenital nephrogenic diabetes insipidus (cNDI) using NanoBiT and cAMP Glosensor assays. The authors show that the mutant proteins have impaired internalization and cAMP responses, and that the OPC5-related agonists are effective in elevating cAMP signalling in some mutants. While the study is generally well written and the findings will be of interest to clinicians and scientists investigating V2R and cNDI, the manuscript needs some amendments to improve clarity. These are all written changes (i.e. no additional experimental work). I have outlined these below.

Revisions

1. The methods are generally very well written with lots of detail. However, I couldn’t see an explanation of how the data presented in the cAMP assays was derived. Presumably this was AUC, but this needs clarifying. This is important for Figure 3 where the authors state the cAMP levels are not affected at early time points but are at later time points. How does this correlate with Fig. 3A? Is the data in 3A reflective of 15 mins or 20 hours? I assume 15 mins (as it is likely AUC of the 30 minute assay), however this needs clarifying.

2. A statistics section should be added to the methods as this is conventional in most publications.

3. The authors state that all mutants have a significant reduction in cell surface expression in Fig. S2. However, no stats are shown. Please add this to the figure.

4. There are several other figures where it is difficult to interpret data as no stats are shown (e.g. Fig. 2B, Fig. 3B-D). These should be added in.

5. In Fig. S2C – in those mutants where the cell surface expression is increased, is this restored to similar levels to wild-type protein or not?

6. Where is the data in Table S2 derived from. I initially thought Fig. 4, however, S2 has N=3 and Fig. 4 N=6. Please clarify this.

7. Figure 4 would be improved by showing wild-type responses to determine whether the compounds restore cAMP levels to wild-type levels.

8. The authors should clarify why they chose 20 hours as the time point to measure later signalling. Is AVP present physiologically for this time period?

9. The authors briefly mention that there are two classes of effects on mutants in the results but then do not expand on this in greater detail in the Discussion. This should be discussed in the context of location of the mutants, degree of inactivation of signalling and other cNDI mutants (e.g. could you predict which mutants are likely to respond). This will make the manuscript of more interest to a wider readership.

10. The authors should clarify why 100nM AVP was used in cAMP assays and 1uM in beta-arrestin assays.

11. The authors state in the discussion that β-arrestin recruitment desensitises the receptor. Although this is the case for many GPCRs, AVPR is known to signal by β-arrestin over long periods of time (via endosomal signaling). The exposure of cells to ligand for 20 hours could affect this endosomal signaling too and this should be discussed in the manuscript.

12. In all the figure legends need to add ‘in’ between ‘performed’ and ‘duplicate’ so that it reads ‘each performed in duplicate.

13. Line 118 – add ‘being’ between ‘before’ and ‘subjected’.

Reviewer #2: In this manuscript, the authors investigate the potential of OPC-51803 (OPC5) and related V2R agonists in addressing partial congenital nephrogenic diabetes insipidus (cNDI), a condition stemming from loss-of-function mutations in the type 2 vasopressin receptor (V2R). The OPC5-related ligands are shown to selectively activate the Gs-cAMP pathway without involving beta-arrestin1/2, suggesting a targeted approach in signaling. The manuscript details the response of six cNDI-related V2R mutants (V882.53M, Y1283.41S, L1614.47P, T2736.37M, S3298.47R, and S3338.51del) to OPC5-related agonists. The manuscript reports a moderate enhancement in cAMP responses in these mutants when treated with OPC5-related ligands compared to AVP, particularly under prolonged stimulation.

Overall the rationale for conducting this comprehensive study is sound. The methods and data analysis also appear to be sound. However, I have concerns about some experiment that seems to be missing and aspects of the experimental techniques used.

1. The authors should assess the activity of OPC analogues in assays directly downstream of Gs coupling with the receptor. This is because the current approach measures one aspect of receptor signaling in close proximity to the receptor (arrestin), while another is evaluated further downstream (cAMP). Such differential measurement depths can result in varying levels of signal amplification. It becomes particularly relevant given the manuscript assertions about biased signaling.

2. Please provide supplementary data using a cell-permeable cAMP analogue (e.g. 8-CPT) to ensure the Glosensor cAMP biosensor is not saturated at maximal ACP ligand concentrations.

3. The authors should investigate the extended cAMP response both in the presence of OPC5 and during its washout (with or w/o antagonist), which could reveal important insights into the kinetics of OPC5, indicating its ability to induce enduring responses.

4. The authors should test whether OPC5 analogues induce differential GRK phosphorylation, in alignment with their profile of biased agonism.

5. The rationale behind measuring receptor internalization at 20 hours and any related observations on receptor recycling to the cell surface need clarification. Assessing total versus cell surface receptors can be achieved through cell lysis.

6. Given that the authors observe cAMP stimulation effects at 20 hours, it’ll be interesting for them to also examine the transcriptional profiling of Gs. Prolonged activation of the cAMP signaling pathway may lead to an amplification of downstream transcriptional responses mediated by Gs.

7. Noting that OPC4 increases receptor expression in some mutants, the ms should also assess its impact on cAMP signaling.

8. In the NanoBiT experiment, the authors have incubated cells with coelenterazine for 2 hours – how is the substrate not burning up and it becomes challenging to differentiate between kinetics of signaling versus kinetics of substrate.

9. Please comment on the activity of OPC5 analogues on the Gq pathway downstream of V2R and its physiological relevance to the disease.

Reviewer #3: In this manuscript, Kuramoto et al. study the cAMP and β-arrestin 1/2 activity of a class of synthetic non-peptide agonists of the vasopressin V2 receptor (V2R). Using in-vitro signaling assays, they show that OPC5 and its analogs generate cAMP responses, albeit much lower than the native peptide agonist AVP, but do not recruit β-arrestin 1/2. Since some V2R mutants are implicated in partial congenital nephrogenic diabetes insipidus (cNDI), the authors compared receptor surface levels and cAMP production by OPC5 analogs in these mutants. Interestingly, OPC5 analogs improved cAMP responses from some of these mutants relative to AVP. These are exciting observations considering the potential for OPC5 analogs to be further developed as therapeutics for partial cNDI. Overall, the manuscript addresses an important question and opens several new directions of future investigation. I have a few suggestions and comments that could be addressed:

1. Fig 1C and D show that treatment with 1μM OPC5 (and its analogs) generate at the most ~10% arrestin recruitment. This assay was performed at a relatively early time point (15 min). On the other hand, receptor cell surface levels were measured after 20 hours of exposure to OPC5 analogs, which is interpreted as receptor internalization (Results, lines 245-246). In these data, there seems to be significant receptor internalization (relative to Veh) for most OPC5 analogs, although clearly not to the extent to which AVP induces internalization.

1a. Do OPC5 analogs induce similar internalization at the early with relatively acute exposure, especially considering previous data show that AVP causes rapid internalization of V2R (https://doi.org/10.1074/jbc.M112.445098;
https://doi.org/10.1091/mbc.e16-12-0818;
https://doi.org/10.7554/eLife.87754.3) ?

1b. Do the authors envision β-arrestin independent internalization with OPC5 analogs at the 20-hour time point?

1c. It might be a good idea to measure whole cell receptor populations to confirm if the reduction in surface receptor levels is due to endocytosis or overall downregulation of receptor levels.

2. Fig S2B: Were the total receptor expression levels (plasma membrane + internal pool) comparable across mutants? Since these experiments were performed in HEK cells, it might be worth discussing on how plasma membrane localization of mutant receptors correlates with data available from endogenous systems.

3. Discussion, line 366: “…non-peptidic agonists may provide a higher therapeutic effect than peptidic agonists on V2R mutants.” Based on the results shown in Fig 4, this seems to be true only in case of Y128S, L161P and T273M mutants.

4. The authors could consider including some more discussion on how specific OPC5 analogs could be potentially applied as biased agonists for cNDI-associated mutants.

5. Placement of significance marks (*s): Fig S2C, S329R: Please check if the ** are correctly placed since OPC16j seems to show a higher cell surface expression relative to Veh, and not OPC19a.

6. Discussion, line 361: please provide references for “greater stability over peptide ligands” for the OPC5 analogs.

7. Introduction: please provide references for V2R signaling described in lines 61-69.

6. PLOS authors have the option to publish the peer review history of their article (what does this mean?). If published, this will include your full peer review and any attached files.

Reviewer #1: **Yes: **Caroline Gorvin

Reviewer #2: No

Reviewer #3: No

---

## [Editor Report · Decision Letter 1]

26 Apr 2024

Therapeutic potentials of nonpeptidic V2R agonists for partial cNDI-causing V2R mutants

PONE-D-23-41336R1

Dear Dr. Inoue,

We’re pleased to inform you that your manuscript has been judged scientifically suitable for publication and will be formally accepted for publication once it meets all outstanding technical requirements.

Kind regards,

Arun Shukla

Academic Editor

PLOS ONE
---

## [Editor Report · Acceptance letter]

2 May 2024

PONE-D-23-41336R1 

PLOS ONE

Dear Dr. Inoue, 

I'm pleased to inform you that your manuscript has been deemed suitable for publication in PLOS ONE. Congratulations! Your manuscript is now being handed over to our production team.

Kind regards, 

on behalf of

Dr Arun Shukla 

Academic Editor

PLOS ONE